# Nonlinear Relationship between Financial Development and CO_2_ Emissions—Based on a PSTR Model

**DOI:** 10.3390/ijerph20010661

**Published:** 2022-12-30

**Authors:** Keyi Duan, Mingyao Cao, Nurhafiza Abdul Kader Malim, Yan Song

**Affiliations:** 1School of Management, Universiti Sains Malaysia, USM, Penang 11800, Malaysia; 2Lee Kuan Yew School of Public Policy, National University of Singapore, Singapore 259772, Singapore; 3Department of City and Regional Planning, University of North Carolina, Chapel Hill, NC 27599-3140, USA

**Keywords:** environmental pollution, CO_2_ emissions, PSTR model, financial development, EKC theory, nonlinear relationship

## Abstract

The contradiction between financial development and environmental pollution has become increasingly prominent with economic development. The discovery of the link between financial development and carbon dioxide emissions will aid in the development of solutions to this problem. This paper uses a panel smooth transition regression (PSTR) model to examine the impact of financial development on carbon dioxide emissions using panel data from 28 Chinese provinces from 2005 to 2021. The PSTR model can solve the problem of minimizing potential outliers ignored in the previous literature, while taking into account the endogeneity and heterogeneity of the model and obtaining more reliable results. According to the findings, financial development has a nonlinear effect on carbon dioxide emissions. Furthermore, the positive effect of financial development on carbon dioxide emissions occurs via the scale and structural effects, while the negative effect occurs via the technological effect, which takes up more space. Moreover, financial added value and the financial scale demonstrate a smooth transition, while financial efficiency and foreign direct investment demonstrate a positive influence.

## 1. Introduction

This study aimed to investigate the nonlinear relationship between financial development and carbon dioxide emissions. With the continuous development of productive forces, economic globalization has been the objective requirement of the development of social productive forces and the inevitable result of scientific and technological progress. Globalization blurs national borders and encourages economic convergence [1], while also affecting and changing human life. This integration increases human demand; all human activities require infrastructure, energy and natural resources for production [2]. Thus, global integration also increases ecological impacts. Greenhouse gas (GHG) emissions represent an alarming issue that causes global warming. The Intergovernmental Panel on Climate Change (IPCC 2021) stated that CO_2_ emissions constitute more than 75% of the total GHG emissions in developing economies. It is a major challenge to reduce the emissions levels for all of the world’s economies [3]. Since the United Nations Conference on Environment and Development in 1992, sustainable development, which is the development of strategies and guidelines for action, has been regarded as a fundamental global development priority, and it has been widely recognized that economic growth and environmental protection must be coordinated and well developed. Although the issue has attracted the attention of many countries, global carbon emissions are still increasing. According to the International Energy Agency (IEA) report (2021), CO_2_ emissions rose by 1.5 billion tons in 2021, with a total of 33 billion tons, reaching a historically high level in the past few years. In some developing countries, environmental problems are more prominent. As shown in Figure 1 and Figure 2, we selected three groups of provinces in China as representatives. Group a included Jilin province, Liaoning province, and Shanxi province, as China’s heavy industry bases. Group b included Jiangsu province, Zhejiang province, and Guangdong province, as China’s fastest-growing provinces with high technological development and high-value-added industries. Group c included Sichuan province, Yunnan province, and Shaanxi province, as the western light industry regions of China.

As is shown in Figure 1a, the CO_2_ emissions of Liaoning province were higher than those in Shanxi province and Jilin province in Group a. The CO_2_ emissions level fluctuated in the period before 2015, and the level then decreased after 2015. As is shown in Figure 1b, in Group b, the CO_2_ emissions for Jiangsu province, Zhejiang province, and Guangdong province decreased from 2007 to 2020. Jiangsu province showed the highest CO_2_ emissions value among the three provinces. Figure 1c indicates that the CO_2_ emissions of Sichuan province and Shaanxi decreased from 2007 to 2020, but the CO_2_ emissions of Yunnan province fluctuated.

As is shown in Figure 2a, the GDP growth of Liaoning province was higher than that of Jilin and Shanxi provinces in Group a, and the GDP of Jilin province was the lowest among the three provinces in Group a. As is shown in Figure 2b, Group b comprised the regions with the fastest-growing GDP in China, which increased year after year; the results showed that Guangdong province had the highest GDP. As is shown in Figure 2c, the GDP of Yunnan and Shaanxi provinces is increasing at a constant rate.

Figure 1 and Figure 2 show that an increase in GDP leads to an increase in carbon dioxide emissions on the whole. For instance, in Group c, Sichuan province showed the greatest reduction in carbon dioxide emissions, as well as the greatest increase in GDP. The more intriguing phenomenon is that carbon dioxide emissions are slightly decreasing. Thus, the relationship between carbon dioxide emissions and economic growth merits further investigation.

The original Kuznets curve (KC) links the country’s development level with inequality. Drawing from KC, Grossman and Kuznets [4] found that there is also an inverted U-shaped relationship between environmental pollution and economic growth. However, economic development is a complex process [5] that will cause the size and structure of the financial sector to change [6]. As is generally agreed, the role of financial development in economic growth and its impact on the environment is critical [7]. Karl, Yuxiang and Zhongchang Chen [8] pointed out that the premise of enterprises in conducting environmentally friendly production is to obtain sufficient financial services. Thus, financial development will improve the environmental quality [9]. However, other scholars hold the opposite view, i.e., that financial development will deteriorate the quality of the environment [10].

The research on financial development and carbon dioxide emissions is at the primary stage. Many researchers have arrived at different results based on EKC theory, and most of them have adopted SDA (Structural Decomposition Analysis), IDA (Index Decomposition Analysis), DEA (Data Envelopment Analysis), GMM (Gaussian mixture model), etc. Some scholars [11,12,13] have adopted SDA to study the relationship between economic development and environmental pollution. However, SDA has an important shortcoming, which is data hysteresis. In addition, it allows cross-period analysis only. This means that SDA can realize a decomposition in the changes in carbon emissions for two base years, which means that it can easily ignore the changes in other years and that it cannot objectively reflect the impact of nonlinear changes between variables on carbon emissions. Some scholars [14,15,16] have used IDA to study carbon emissions in China. Although IDA can favorably realize the differential expansion of various explanatory variables, the problem of decomposition residuals still remains. Additionally, some scholars [17,18,19] have used a DEA model to study the correlation between economic growth and energy consumption. Undoubtedly, DEA models can cope with multiple input and output issues, and they are free of the impact of different scales and can deal with interval data as well as ordinal data. However, they cannot be used to conduct further assessments of variables. The relative results of panel data cannot be compared. In addition, DEA models need a large sample size; otherwise, the results might be unreliable.

Most of the existing studies consider the direct impact, or only use a single set of financial development data to examine the nonlinear impact of financial development on carbon emissions. A series of empirical studies has verified the EKC [20,21,22]. However, there are few studies on the impact of financial development on China’s carbon emissions. In view of the inconsistency in the literature and these knowledge gaps, in this study, we performed a comprehensive evaluation of the impact of financial development on carbon dioxide emissions, using multiple indicators. Most conclusions assume that the correlation between carbon emissions and the economic growth of countries at different economic development stages is homogeneous. As a result, the differences in the economic structure, cultural traditions and historical development of different countries are ignored. The conclusions thus obtained are not applicable to any specific country. Furthermore, these conclusions cannot be converted into constructive suggestions for specific countries.

The rest of this paper is organized as follows. The Section 2 presents the literature review, the Section 3 describes the methodological model and data, the Section 4 presents the empirical results, and the Section 5 presents the conclusions and policy recommendations.

## 2. Literature Review

Numerous studies have attempted to describe the relationship between financial development and CO_2_ emissions. For example, as shown in Table 1, Shahbaz, Pradeepta, Li, Eyup, James, Jungbo, Adnan, Kazem and Jordi [20,23,24,25,26,27,28,29,30] confirmed the nexus of financial development and carbon dioxide emissions in South Africa, India, EU countries, China, European Countries, France, Nordic Countries and Spain. There are two conflicting opinions about the relationship between financial development and carbon dioxide emissions. The research by Tamazian et al. [31] indicated that carbon dioxide emissions decrease with the constant development of finance. In other words, financial development can curb carbon dioxide emissions. Shahbaz and Sinha [32] empirically analyzed relevant data from France to verify the EKC curve and found that financial development can lead to a reduction in carbon emissions, which can help to improve the environmental governance of France. Additionally, they suggested that financial stability is a necessary condition for an improvement in the quality of the environment. Jalil and Feridun [7] further studied the correlation between indices such as financial development, economic growth, energy consumption and environmental pollution. The research results implied inverse changes in financial development and carbon dioxide emissions, meaning that financial development and environmental pollution are negatively correlated [8,33]. However, Allen and Yago [34] argued that it was necessary to make full use of enterprises’ scarce resources, reduce enterprises’ financing costs and clarify that economic growth is mainly responsible for enterprises’ environmental pollution. Muhammad and Khan [35] used empirical data to verify the mutual impact among financial development, economic growth, energy consumption and carbon emissions. With the worsening of pollution, energy consumption can curb economic growth. In contrast, Bello and Abimmbola [36] focused on Nigeria to conduct a case study to examine the impact of the country’s financial development on environmental pollution. The results show that due to financial departments’ lack of supervision of enterprises’ borrowing for investment, the positive flow and efficient utilization of funds cannot be ensured. Therefore, financial development can, to some extent, accelerate the worsening of the environment in the long run. Duan et al. [37] used PVAR model to find that there is a two-way causal relationship between financial development and carbon dioxide emissions, they also agree financial development has a negative impact on environment. Zaidi et al. [38] established the dynamic linkages between globalization, financial development and carbon dioxide emissions, and they highlighted that there is a feedback effect between these two nexuses.

According to Charfeddine and Khediri [39], financial sector development is fundamental in increasing economic growth, and it also attracts more foreign direct investment. Therefore, we need to take into account the impact of foreign direct investment on the environment. Shahbaz et al. [40] found that financial development will increase FDI, which results in more energy consumption. Grimes and Kentor [41] used panel data covering developing countries from 1980 to 1996 to study the correlation between FDI and carbon emissions. Their research findings suggested that FDI and carbon emissions are significantly and negatively correlated. Based on data from multiple countries in 1999, Prew [42] verified the EKC model, with the results suggesting that the EKC model can be substantiated. In other words, FDI can weaken the pollution discharge of host countries. Similarly, Pao and Tsai [43] used relevant data from BRIC (Brazil, Russia, India and China) countries from 1980 to 2007 and studied the correlation between FDI and the environmental pollution of host countries using the steady test, cointegration test, ordinary least squares (OLS) and other methods. Their research findings also significantly support the EKC hypothesis, which means that FDI can increase the carbon emissions of host countries. The dynamic correlation between FDI and the carbon emissions of host countries is taken into consideration.

We focus on financial development in a more comprehensive manner by including foreign direct investment, financial added value and financial scales in our analysis. In this context, we analyze the nonlinear relationship between financial development and carbon dioxide emissions for China.

**Table 1 ijerph-20-00661-t001:** Survey of different research studying the relationship between financial development and carbon dioxide emissions.

**Author**	**Variables**	**Method**	**Conclusion**	**Countries**
Shahbaz [20]	CO_2_, FD, EGY, EP, Y	ADL, Ng-perron unit root	EGY and Y contribute to CO_2_; TR and FD mitigate CO_2_	South Africa
Pradeepta Sethi [29]	CO_2_, CE, GI, FD, EG, U, EGY	ARDL	FD contributes to EGY, CE and GI; U mitigates EN	India
Li [28]	EGY, TR, CO_2_	GMM, Mean Group Estimation	EGY and TR contribute to CO_2_	China
Eyup [27]	TR, FD, NREC, REC	CADF, CIPS, Heterogeneous Panel	TR, FD and REC mitigate CO_2_; NREC contributes to CO_2_	European Countries
Wang [44]	UR, CE	Panel root test	UR contribute to CO_2_	BRICS countries
James B [23]	CO_2_, EGY, Y	ADF, ARDL Model	EGY contributes to CO_2_	France
Jungho [25]	CO_2_, FD, EYG	ARDL Model	FD contributes to CO_2_; EYG negatively affects CO_2_	Nordic Countries
Adnan [26]	CO_2_, EGY, Y, TR, GDP	Panel Unit Root Tests, Panel Cointegration Method and Panel Causality Test	EGY, TR and URB contribute to CO_2_	EU member
Kazem [30]	CO_2_, Nitrogen Oxide, Carbon Monoxide, Sulfur Dioxide, Sulfur Trioxide, GDP	Panel Unit Root Test, Cointegration Test, DOLS	GDP contributes to CO_2_, nitrogen oxide, carbon monoxide, sulfur dioxide and sulfur trioxide	Iran’s provinces
Jordi [24]	GDP, Gas Emissions, T	Input–Output Approach, SDA	T mitigates CO_2_; GDP contributes to CO_2_	Spain

NOTE: FD (financial development), Y (income), EP (environmental pollution), ADL (autoregressive distributed lag model), ARDL (autoregressive distributed bound test), CE (environmental degradation), GI (globalization index), U (urbanization), SLM (static spatial panel model), IS (industrial structure), T (technology), TR (trade), REC (renewable energy consumption), NREC (nonrenewable energy consumption), EYG (aggregate energy consumption), DOLS (dynamic ordinary least squares), SDA (structural decomposition analysis), UR (share of urban population, CE (carbon dioxide emission per capita.

## 3. Model Setting and Data Description

### 3.1. Panel Smooth Transition Regression (PSTR) Model

In a complex economic system, different variables are usually nonlinearly correlated. Among models that are used to portray nonlinear characteristics in economic phenomena, the regime switching model, which is convenient and easy to use, can well explain the periodic and asymmetric characteristics in economic phenomena.

Based on different hypotheses about regime switching behaviors, regime switching models can be divided into three categories, namely the smooth transition model, threshold model and Markov regime switching model. A striking characteristic of the panel smooth transition regression (PSTR) model is that it allows its regression coefficient to gradually change when gradually moving from one group to another group. In other words, the parameters in the model can realize continuous and gradual smooth transitions between different extreme regimes as one function containing exogenous variables. This characteristic is in line with the characteristics of economic development. As stated by Gainelli et al. [45], PSTR is beneficial for minimizing the effect of potential outliers, allowing researchers to obtain findings that are free of any outliers. PSTR can also endogenously predict the threshold level and estimate the degree of excessive smoothing from low-income areas to high-income areas. Unlike traditional methods, a PSTR model can take into account endogeneity and heterogeneity. Additionally, linear tests and residual nonlinear tests are carried out, which can ensure the higher stability and reliability of the test results. For example, Ulucak [46] used a PSTR model to study the nonlinear impact of globalization on material consumption in EU countries. Hu [47] used a PSTR model to study the nonlinear relationship between the ownership structure and optimal capital structure. PSTR was also used to analyze a large panel of data covering 146 economies to verify the relationship between economic growth and CO_2_ emissions [48]. It is undoubtedly a more flexible method for the investigation of cross-country heterogeneity and time instability. PSTR models are also used in researching the patent–growth relationship [29], liquidity growth [49] and environmental regulation [50]. Since we study the relationship between economic growth and carbon dioxide emissions, the economic changes are not dramatic. Instead, these changes happen gradually along with the changes in financial added value, the financial scale, financial efficiency, technology and foreign direct investment (FDI). Considering that the effect of economic growth on the environment is nonlinear, we adopt a PSTR model and conduct research under a nonlinear framework.

To better explain the method adopted in this study, the details of PSTR are as follows.

By replacing the discrete transition function in the PTR with a continuous transition function, the model coefficients can vary continuously with the transition variables. PSTR is a panel data model that adds a continuous transfer function to the model that is more in line with socioeconomic reality and addresses the issue of a discontinuous leap in the threshold value in Hansen’s PTR model more effectively. The PSTR model, which is appropriate for multi-section panel data research, has the advantage of successfully capturing cross-section variability in the analytical process as well as smooth transformation. The general form of the simple two-system PSTR model is depicted in Equation (1):(1)$$y_{it}=\mu_{i}+\beta_{1}x_{it}+\beta_{2}x_{it}g(q_{it};\gamma ,c)+\varepsilon_{i}$$

According to Gonzale and Vijik, $g\left(q_{it};\gamma ,c\right)$ usually adopts the following logical function form:(2)$$g(q_{it};\gamma ,c)={\left(1+\exp(-\gamma \prod_{j=1}^{m}(q_{it}-c_{j}))\right)}^{-1},c_{1}\leq c_{1}\leq ...\leq c_{m},\gamma >0$$

When the transition function is present, the corresponding Equation (2) is referred to as a low system, and when the transition function is present, as a high system. Equation (2) may seamlessly switch between a high system and a low system because the transition function’s value changes between 0 and 1 gradually. The PTR model then replaces the PSTR model if.

The smooth transition model (STR) and the panel transition model are both extensions of the panel smooth transition regression (PSTR) model (PTR). Equation (3) illustrates them ultisystem PSTR model’s general form.
(3)$$y_{it}=\mu_{i}+\beta_{1}x_{it}+\sum_{j=1}^{\gamma }\alpha_{j}x_{it}g_{j}(q_{it}^{j};\gamma ,c)+\varepsilon_{i}$$

$\mu_{i}$ is the individual effect, $\beta_{1}$ is the regression coefficient of the linear part of the explanatory variable, c is the location parameter, $x_{it}$ is the explanatory variable, and $\varepsilon_{i}$ is the random error term, where I = 1, 2, …, N. T = 1, 2, …, T is a transition function that changes continuously between 0 and 1. The observable value $q_{it}^{j}$ is the transition variable, and γ is a smoothing parameter that determines the transition speed. *c* is the location parameter at which the transition occurs.

As a prerequisite for the PSTR model to be estimated when characterized by nonlinear effects, in the relevant literature, nonlinear effects are generally tested using asymptotic values of statistics, such as the LM, LMF and LRT statistics. It should be noted that before the above LM, LMF and LRT tests can be used to reject the null hypothesis H0: r=0, it is necessary to further test for residual nonlinear effects (i.e., test the number of transition functions). Ulucak et al. [46] noted that, in general, m is the number of position parameters, and the PSTR model is able to satisfy the m = 1 or m = 2 cases considered by the researchers through 1–2 parameter transformations. Nonetheless, the specific value of m is still to be tested, and the null hypotheses $H_{01}^{*}:\beta_{1}^{*}=0|\beta_{2}^{*}=\beta_{3}^{*}=0\;H_{02}^{*}:\beta_{2}^{*}=0|\beta_{3}^{*}=0\;H_{03}^{*}:\beta_{3}^{*}=0\;$ should also be tested in turn. The m value can be determined when the degree of rejection is the strongest.

In this work, the financial development variable is considered an excessive variable, and the specific equation is as shown in Equation (4). In Equation (4), CO_2_ indicates the economic level, Y indicates the economic level, S indicates the industrial structure, T indicates the technical level, FD_1_ indicates the financial added value, FD_2_ indicates the financial scale, FD_3_ indicates the financial efficiency and FD_4_ indicates foreign direct investment.
(4)$${\mathrm{lnCO}}_{2}=\alpha_{1,1}\ln Y_{it}+\alpha_{1,2}\ln S_{it}+\alpha_{1,3}\ln T_{it}+\alpha_{1,4}\ln DF_{it}+(\alpha_{2,1}\ln Y_{it}+\alpha_{2,2}\ln S_{it}+\alpha_{2,3}\ln T_{it}+\alpha_{2,4}\ln FD_{it})g(\ln FD_{it};\gamma ,c)+\mu_{it}CO_{2}$$

### 3.2. Data Description

As shown in Table 2, this study uses 30 Chinese provincial administrative units as observations, and panel data from 2005 to 2021 to measure the specific variables. Equation (5) was used to evaluate the CO_2_.
(5)$${\mathrm{CO}}_{2}=\sum \alpha_{i}\beta_{i}E_{i}$$

*α_i_* denotes to the standard coal factor of energy source *i*, *β_i_* denotes CO_2_ emission factor e, and *E_i_* denotes the consumption of energy source *i α_i_* and *β_i_* are mainly taken from the IPCC and the China Statistical Yearbook published by each provincial government in China (Table 3). In addition, the data of other variables are derived from the China Statistical Yearbook, and Table 2 provides the data used in this study.

**Table 2 ijerph-20-00661-t002:** Variables Setting.

Symbols	Variables	Definitions
CO_2_	Carbon dioxide emission	Total carbon dioxide emission
Y	Economic level	Gross regional product
S	Industrial structure	The ratio of the added value of the secondary industry to regional GDP
T	Technical level	Number of authorized patent applications (items)
FD1	Financial added value	The ratio of financial added value to regional GDP
FD2	Financial scale	The ratio of balance of deposits and loans of financial institutions to regional GDP
FD3	Financial efficiency	The ratio of loan balance to deposit balance of a financial institution
FD4	Foreign direct investment	The ratio of foreign direct investment to regional GDP

## 4. Results and Discussion

### 4.1. Descriptive Statistics Analysis

Table 4 shows the descriptive statistics. As is shown in Table 4, the observed values for lnCO_2_, lnY, lnS, lnT, lnFD_1_, lnFD_2_, lnFD_3_ and lnFD_4_ were 900. The average values for these parameters were 9.432, 14.869, −0.762, 9.028, −3.096, 19.336, −0.236, −1.964, respectively. The standard deviations for these parameters were low. In addition, Table 3 also shows the minimum and maximum values.

### 4.2. Linear Test

The linear test between financial growth and carbon dioxide emissions is displayed in Table 5. We reject the null hypothesis that changes in the financial sector have a linear effect on carbon dioxide emissions. We discover that there is a large nonlinear link between financial development and carbon dioxide emissions. The initial hypothesis is rejected at the 1% level of significance in Model 1 through Model 4, with m changing from one to five. The parameters’ positions are represented by the values of m, which is further decided by the AIC and BIC’s unofficial rules. As was shown in Table 6, different models showed the different location parameters.

### 4.3. Residual Nonlinear Test

The findings of the residual nonlinearity test are displayed in Table 7 and Table 8. Using Mosikari and Eita (2020) and other sources as our foundation [49], we create the cases with the position parameter m = 1, m = 2, and m = 3 and test the associated cases up until the direct model accepts the null hypothesis. The outcomes of Model 1 demonstrate the acceptance of the null hypothesis H0: r = 1. There is just one transition function, and r = 1 is confirmed. Models 2, 3, and 4 all provide the same results. In these four models, there is only one transition function.

Table 9 displays the relationships between financial growth and CO_2_ emissions for Models 1 and 2. In Model 1, the relationship between financial value added and CO_2_ emissions has a linear coefficient value of 0.4824 and a nonlinear coefficient value of 0.0842, with a value of at least 48.24% and a maximum of 56.66% (0.4824 + 0.0842 = 0.5666). Financial value added is a measure of financial development. These findings imply that economic expansion reduces CO_2_ emissions through a catalytic process that raises the added value of the economy. We also provide research support for the hypothesis that financial development increases CO_2_ emissions through the scale effect (Zhang, 2011). (Zhang, 2011). Additionally, the model’s lnS linear and nonlinear coefficients are 0.3311 and 1.2134, respectively. Furthermore, the industrial structure contributes to CO_2_ emissions from 33.11% to 154.45% (0.3311 + 1.2134 = 1.5445). These findings suggest a more pronounced impact of capital on carbon dioxide emissions. Therefore, capital flows must be strategically managed to prevent excessive capital flows to secondary industries, which might have negative effects on the environment. In the current scenario in China, the nation is in the process of industrial structure optimization and upgrading, and controlling financial capital allocation may effectively enable China to optimize its industrial structure.

In contrast, the linear coefficient of lnT is equal to −0.1128, and its nonlinear coefficient is equal to −0.2436, which shows that technological advancement typically has a limiting effect on CO_2_ emissions between 11.28% and 35.64%. So, via the impact of technology, financial development may lower CO_2_ emissions. Further study reveals that China has a wider space to sustainably cut CO_2_ emissions through technological progress. In addition, the coefficient of lnFD is 0.0846, and the nonlinear coefficient is 0.1169 (not significant), which reveals the nonlinear influence of financial development on CO_2_ emissions, namely from 0.0846 to 0.2015 (0.0846 + 0.1169 = 0.2015). The higher scale effect and structural effect relative to the technological effect may account for the significant variation in the threshold. According to the thorough study, a suitable steering system should be set up to direct financial resources in order to assist technical growth and enhance the technological effect, resulting in financial development for CO_2_ emission reduction. In reality, the smoothness value of Model 1 is 1524, which suggests that the transition process is quite quick.

A measure of financial development, the financial scale in Model 2 has a linear coefficient of 0.3613 and a nonlinear coefficient of 0.2531, ranging from 36.13% to 61.44% (0.3613 + 0.2531 = 0.6144), which together make up the influence of the financial scale on CO_2_ emissions. We offer evidence in favor of the hypothesis that scale-effect CO_2_ emissions can be increased by financial development. Industrial restructuring can effectively contribute to CO_2_ emissions, with an effect that ranges from 13.25% to 104.9%, according to the linear coefficient value of lnS, which is equal to 0.1325, and the nonlinear coefficient, which is equal to 0.9165.LnT has a nonlinear value of 0.3824 and a linear value of −0.3625 A more thorough examination reveals that when the amount of money is utilized as a transition variable and is followed by a suppression impact of 36.25% and a boosting effect of 1.99%, the direction of the function at the technology level is not the same. This outcome might be partially attributed to the growth in the amount of funding allocated for R&D in non-low-carbon technology. Financial development has an influence on CO_2_ emissions, ranging from a suppressive effect of 53.89% (0.1124 + 0.4265 = 0.5389) to a facilitative effect of 11.24%, according to the linear coefficient of lnFD of 0.1124 and the nonlinear coefficient of 0.4265. Because of this, financial development may reduce CO_2_ emissions as much as possible by controlling the flow of money. Model 2 has a relatively modest smoothing parameter of 1568, which causes the transition process to be gradual and persistent.

Table 10 displays the relationships between financial growth and CO_2_ emissions for Models 3 and 4. Financial development factors in Model 3 are measured using financial efficiency. The results of Models 1 and 2 are consistent with the linear coefficient of lnY being equal to 0.4268 and the nonlinear coefficient being equal to 0.1387, which indicates that the impact of economic growth on CO_2_ emissions is between 42.68% and 56.55% (0.4268 + 0.1387). Since the nonlinear coefficient of lnS is 0.2946 and the linear coefficient of lnS is 0.3316, the industrial structure contributes between 33.16% and 62.62% of the CO_2_ emissions (0.3316 + 0.2946 = 0.6262). This finding implies that the growth of secondary sector has the potential to considerably raise CO_2_ emissions. According to the technological level, the inhibitory impact on CO_2_ emissions ranges from 10.62% to 11.72% (−0.2234 + 0.1062 = −0.1172), with a linear value of lnT equal to −0.2234 and a nonlinear coefficient of 0.1062. Financial efficiency contributes positively to CO_2_ emissions with a range of 23.26% to 56.52%, as shown by the linear coefficient value of lnFD of 0.2326 and the nonlinear coefficient of 0.3326 (0.2326 + 0.3326 = 0.5652). Model 3 demonstrates that financial development has both benefits and drawbacks. As a result, we must address the negative consequences of the expansion of the economy and concentrate on the rise in CO_2_ emissions brought on by economic growth.

As a way of measuring the financial development variable in Model 4, we employ foreign direct investment (FDI). Indicating that the impact of economic growth on CO_2_ emissions ranges from 23.54% to 36.95% (0.2354 + 0.1341 = 0.3695), the linear coefficient of lnY is equal to 0.2354, and the nonlinear coefficient is equal to 0.1341. These results show that both the scale effect and the effects of economic growth on CO_2_ emissions are strong. The nonlinear coefficient of lnS is equal to 1.2216, and the linear coefficient of lnS is equal to 0.3214, demonstrating that the industrial structure has a large promoting impact on CO_2_ emissions with a wide range of influence spanning from 32.14% to 154.3%. Technology level has an inhibitory influence on CO_2_ emissions between 5.04% and 18.32%, as shown by the linear coefficient of lnT, which is equal to −0.1328, and the nonlinear coefficient, which is 0.1832 (−0.1328 + 0.1832 = 5.04). According to the linear coefficient of lnFD, which is equal to 0.2348, and the nonlinear coefficient, which is 0.2645 (not significant), financial development, as measured by FDI, also adds to CO_2_ emissions, which range from 23.48% to 49.93%. In conclusion, FDI makes a small difference in CO_2_ emissions compared to the development of the economy and the structure of the industrial sector. These findings are in line with Zhang’s (2011) research, which indicates that FDI has little impact on GDP and has no discernible influence on CO_2_ emissions.

### 4.4. The Contribution of This Research

This work uses financial added value, the financial scale, financial efficiency and foreign direct investment to represent financial development. In the literature on financial development, the financial scale (the ratio of balance of deposits and loans to regional GDP) has the advantage of “easy-to-obtain data” and is most extensively used; therefore, this ratio often used as an indicator of financial development [51,52]. This study follows the method of Wang et al. [53] for reference and adopts the ratio of the balance of deposits and loans of financial institutions to regional GDP (financial scale) and the ratio of the loan balance to the deposit balance of financial institutions (financial efficiency), respectively, to reflect financial development. The ratio of financial added value to regional GDP is estimated based on the method of Kong and Wei [54].

The main contribution of this work is as follows.

This study is different from previous research, such as Abdul et al. [55], which also tested EKC theory, finding that there is a unidirectional relationship between globalization, financial development and carbon dioxide emissions; in their work, the data are heterogeneous, and the result is not reliable. The present study is distinguished from studies that consider only the inconsistent linear relationship between financial development and carbon dioxide emissions, and it develops a new method (panel smooth transition regression, PSTR) to address previously unresolved problems, such as potential outliers.

In addition, this study takes one country for analysis and introduces variables, including financial added value, the financial scale, financial efficiency and foreign direct investment, to examine the correlation between economic development and environmental pollution. This can avoid the limitations of many other researchers who adopt samples from different countries, such as EU countries [26], African countries [20] and Arctic countries [25]. These countries follow different energy policies, which makes it difficult to identify whether EKC theory is applicable to every country, particularly countries that are at the same economic development level but that adopt different paths for environmental protection.

Lastly, carbon dioxide is a major contributor to the greenhouse effect. In this work, carbon dioxide, which has a stronger spatial spillover effect, is used as an index to measure environmental pollution. Stern [56] pointed out that the emissions of greenhouse gases are the largest market failure that mankind has ever encountered. It is hoped that some policy suggestions can be made by analyzing the environmental problems resulting from China’s economic development to fill the gaps in existing research.

## 5. Conclusions

We provide novel evidence of the nonlinear effect of financial development on carbon dioxide emissions and explore the smooth transition mechanism based on the PSTR model, using panel data covering 28 provinces in China from 2005 to 2021. This study’s main conclusions were as follows:

Financial development has a nonlinear impact on carbon dioxide emissions. Financial development can increase CO_2_ emissions through added value and the scale effect. The financial scale effect has a positive effect on CO_2_ emissions from 11.24% to 54.89%; thus, under the scale effect, the coefficient value ranges from 48.24% to 56.66%, and 36.13% to 61.44%, respectively.

Financial development is also measured by new indices—for example, financial added value, the financial scale, financial efficiency and foreign direct investment. The first two variables present a smooth transition impact on carbon dioxide emissions; however, the last two variables always show a promoting effect. The contribution of FDI to carbon dioxide emissions ranges from 23.48% to 49.93%. Financial development facilitates an improvement in financial added value through the scale effect, thus accelerating carbon dioxide emissions.

We also make some suggestions for the mitigation of carbon dioxide emissions.

1. The production of industry relies on the emission of polluting gases. Enterprises need to increase their technological innovation in production equipment, improve their production efficiency, realize high-tech production and reduce environmental pollution; the government can increase its financial support for green industries and gradually eliminate high-tech polluting enterprises, while monitoring the disclosure of corporate environmental performance information, providing strict supervision and helping enterprises to achieve green transformation. In addition, the use of clean energy and renewable energy can improve infrastructure construction. Moreover, China’s government should optimize the industrial structure and promote industrial upgrading. It is necessary to develop an outline and plan for the development of emerging strategic industries, use the leapfrog development of emerging strategic industries to promote industrial structure upgrades, change the industrial system featuring high energy consumption and high emissions to low-energy-consumption and high-value-added industries and promote economic transformation.

2. It is necessary to invest in the research and development of low-carbon technologies to accelerate the transition to high-value-added manufacturing and low-energy-consumption products. The results of the study show that financial development can reduce CO_2_ emissions through the technological effect. The improvement of the quality of China’s economic growth depends on the growth of innovation input. On the basis of the improvement of innovation input, the diversification of urban innovation input should be improved to promote the spillover effect of inter-industrial innovation activities and support the improvement of the overall quality of urban economic development. With the upgrading and adjustment of China’s manufacturing industry, it is possible to attract domestic and foreign R&D enterprises and institutions in different industries, especially in order to promote the technology spillover effect of foreign investment. Enterprises should strengthen the formal and informal innovation exchanges between foreign-funded enterprises and domestic-funded enterprises and achieve growth in innovation efficiency by building effective inter-industrial cooperation, thus effectively improving the quality of China’s urban economic growth.

3. To facilitate the relevant aspects of sustainable development, it is recommended that FDI flows be directed into more sustainable and greener industries of the economy. In terms of financial development, it is also critical to allocate financial resources to environmentally friendly sectors of the economy; thus, green finance is the way forward for China’s integration.

## Figures and Tables

**Figure 1 ijerph-20-00661-f001:**
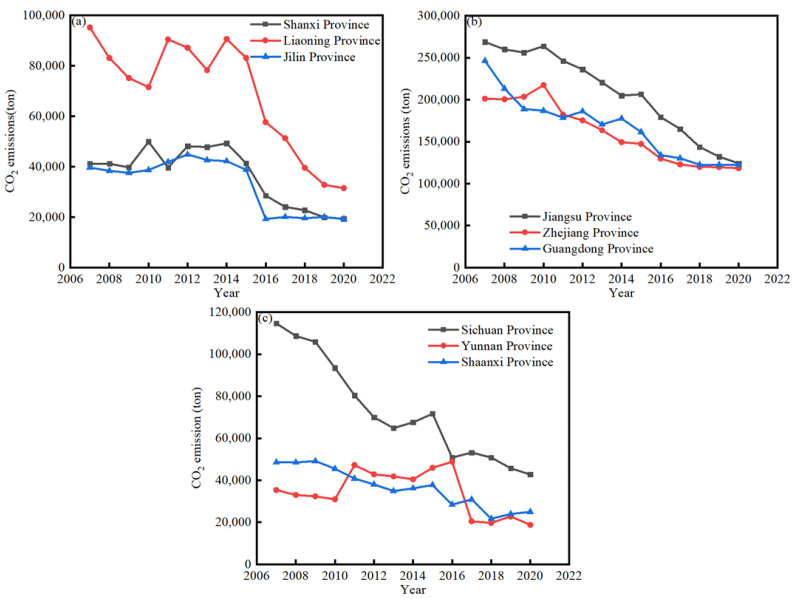
(**a**) Shanxi, Liaoning, and Jilin Provinces; (**b**) Jiangsu, Zhejiang, and Guangdong Provinces; (**c**) Sichuan, Yunnan, and Shaanxi Provinces. Change in CO_2_ emissions over time (2006–2020) (Data: IBRD. IDA).

**Figure 2 ijerph-20-00661-f002:**
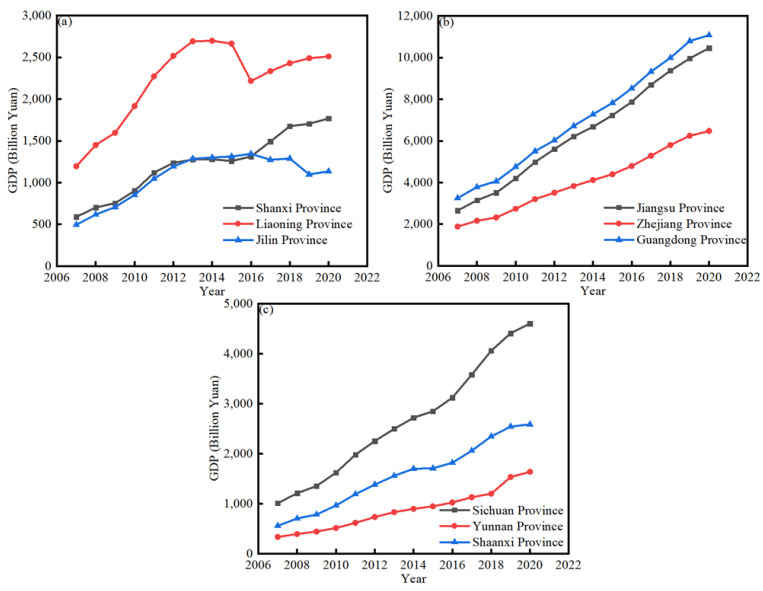
(**a**) Shanxi, Liaoning, Jilin and Provinces; (**b**) Jiangsu, Zhejiang, Guangdong and Provinces; (**c**) Sichuan, Yunnan, and Shaanxi Provinces. Change in GDP over time (2006–2020) (Data: IBRD. IDA).

**Table 3 ijerph-20-00661-t003:** Energy standard coal coefficient *α_i_* and carbon dioxide emission coefficient *β_i_*.

Energy	Oil Sands	Coke	Crude Oil	Fuel	Gasoline	Diesel	Natural Gas
αi	0.385	0.756	0.325	0.426	0.038	0.534	0.246
βi	0.846	1.073	1.124	1.365	1.075	1.864	0.815

**Table 4 ijerph-20-00661-t004:** Descriptive statistics.

Variables	Observed Values	Average	Standard Deviation	Minimum	Maximum
lnCO_2_	900	9.432	0.369	5.148	7.221
lnY	900	14.869	0.772	14.268	19.381
lnS	900	−0.762	0.254	−1.625	−0.488
lnT	900	9.028	1.472	5.302	15.148
lnFD_1_	900	−3.096	0.462	−5.036	−0.752
lnFD_2_	900	19.336	1.248	13.192	17.068
lnFD_3_	900	−0.236	0.328	−0.558	0.407
lnFD_4_	900	−1.964	0.826	−2.549	1.418

**Table 5 ijerph-20-00661-t005:** The linear test.

	Model 1		Model 2		Model 3		Model 4	
	F	*p*-Value	F	*p*-Value	F	*p*-Value	F	*p*-Value
m = 1	11.32	0.004	11.28	0.007	7.48	0.002	22.38	0.009
m = 2	14.63	0.007	32.54	0.007	13.48	0.009	13.45	0.008
m = 3	13.78	0.008	34.12	0.012	10.85	0.008	13.46	0.018
m = 4	14.98	0.008	34.16	0.009	13.38	0.007	18.85	0.013
m = 5	16.11	0.006	34.54	0.008	15.89	0.003	18.94	0.008

**Table 6 ijerph-20-00661-t006:** Location parameters.

	Model 1		Model 2		Model 3		Model 4	
	AIC	BIC	AIC	BIC	AIC	BIC	AIC	BIC
m = 1	−3.261	−3.424	−2.864	−2.584	−3.846	−4.122	−3.426	−3.654
m = 2	−4.124	−3.652	−2.896	−3.984	−3.462	−2.648	−2.948	−2.584
m = 3	−3.264	−3.784	−3.462	−3.682	−3.644	−3.462	−3.861	−2.984
m = 4	−4.019	−3.451	−3.826	−3.127	−4.126	−2.856	−3.084	−3.542
m = 5	−2.894	−3.214	−3.024	−4.126	−4.246	−3.012	−3.864	−3.358

**Table 7 ijerph-20-00661-t007:** Residual nonlinear test.

		Model 1		Model 2	
		F	*p*-Value	F	*p*-Value
m = 1	H0: r = 0 vs. H1: r ≥ 1	14.28	0.012	9.88	0.006
	H0: r = 1 vs. H1: r ≥ 2	22.76	0.068	22.34	0.158
m = 2	H0: r = 0 vs. H1: r ≥ 1	13.68	0.014	33.58	0.024
	H0: r = 1 vs. H1: r ≥ 2	4.68	0.462	3.942	0.087
m = 3	H0: r = 0 vs. H1: r ≥ 1	12.09	0.006	33.09	0.006
	H0: r = 1 vs. H1: r ≥ 2	7.28	0.084	17.65	0.109
m = 4	H0: r = 0 vs. H1:r ≥ 1	10.59	0.008	21.49	0.013
	H0: r = 1 vs. H1: r ≥ 2	7.08	0.259	16.08	0.018
m = 5	H0: r = 0 vs. H1: r ≥ 1	11.21	0.006	12.48	0.015
	H0: r = 1 vs. H1: r ≥ 2	5.24	0.084	13.58	0.017

**Table 8 ijerph-20-00661-t008:** Residual nonlinear test.

		Model 3		Model 4	
		F	*p*-Value	F	*p*-Value
m = 1	H0: r = 0 vs. H1: r ≥ 1	5.88	0.014	22.21	0.009
	H0: r = 1 vs. H1: r ≥ 2	6.89	0.184	3.59	0.048
m = 2	H0: r = 0 vs. H1: r ≥ 1	13.15	0.002	12.84	0.007
	H0: r = 1 vs. H1: r ≥ 2	5.84	0.068	2.16	0.028
m = 3	H0: r = 0 vs. H1: r ≥ 1	9.58	0.003	13.14	0.023
	H0: r = 1 vs. H1: r ≥ 2	3.02	0.254	2.46	0.022
m = 4	H0: r = 0 vs. H1: r ≥ 1	9.14	0.008	13.38	0.046
	H0: r = 1 vs. H1: r ≥ 2	4.12	0.084	4.68	0.054
m = 5	H0: r = 0 vs. H1: r ≥ 1	8.22	0.012	11.64	0.126
	H0: r = 1 vs. H1: r ≥ 2	6.12	0.258	2.48	0.214

**Table 9 ijerph-20-00661-t009:** Estimated results.

Variables	Model 1		Model 2	
lnCO_2_	Linear Part	Nonlinear Part	Linear Part	Nonlinear Part
lnY	0.4824 **	0.0842 **	0.3613 **	0.2531 ***
	(18.3128)	(2.5714)	(3.2214)	(2.3412)
lnS	0.3311 **	1.2134 ***	0.1325 **	0.9165 ***
	(3.1206)	(10.5437)	(0.8438)	(5.0234)
lnT	−0.1128 **	−0.2436 **	−0.3625 **	0.3824 **
	(−3.1244)	(−3.5842)	(−6.2142)	(5.3176)
lnFD	0.0846 **	0.1169 ***	0.1124 **	0.4265 ***
	(−0.3164)	(1.5736)	(2.1256)	(−2.8456)
Location parameters	−3.1254		16.2138	
	−2.1245		18.1268	
Smoothing parameters	1524		1568	

** 1.964. *** 2.584.

**Table 10 ijerph-20-00661-t010:** Estimated results.

Variables	Model 3		Model 4	
lnCO_2_	Linear Part	Nonlinear Part	Linear Part	Nonlinear Part
lnY	0.4268 **	0.1387 ***	0.2354 **	0.1341 **
	(17.5463)	(1.029)	(20.2238)	(2.8426)
lnS	0.3316 ***	0.2946 ***	0.3214 ***	1.2216 **
	(6.2168)	(3.3452)	(4.0318)	(7.5838)
lnT	−0.2234 ***	0.1062 ***	−0.1328 **	0.1832 **
	(2.1826)	(1.1386)	(−2.8564)	(0.0846)
lnFD	0.2326 ***	0.3326 **	0.2348 **	0.2645 ***
	(1.2369)	(4.2836)	(4.1623)	(0.1268)
Location parameters	0.2462 **	0.1826 ***	0.1158 **	0.1116 ***
	(0.8836)	(3.0814)	(2.1361)	(0.4216)
Smoothing parameters	46.2159		15.2168	

Note: ** *p* < 0.05, *** *p* < 0.01.

## Data Availability

Not applicable.

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
