# Peer review of "Nonlinear Relationship between Financial Development and CO2 Emissions—Based on a PSTR Model"

_ijerph, 2022, doi:10.3390/ijerph20010661_

Round 1

Reviewer 1 Report (Previous Reviewer 2)

I suggest that Section3.3 be placed in the discussion section and the title of Section 3.3 be appropriately modified.

Author Response

Thanks for this meaning suggestion. Section 3.3 was removed to discussion section, which was shown in Line 488 to Line 521. We also changed the title to “The contribution of this research”.

Reviewer 2 Report (Previous Reviewer 3)

The paper has been significantly improved and the overall merit has increased.

I would advise for publication.

Author Response

Thank you very much for your feedback and approval for our manuscript.

Reviewer 3 Report (Previous Reviewer 5)

I think that the revised manuscript addresses all my previous concerns and comments. Congrats to the authors. 

Author Response

Thank you very much for taking the time to review and accept our manuscript.

This manuscript is a resubmission of an earlier submission. The following is a list of the peer review reports and author responses from that submission.

Round 1

Reviewer 1 Report

Research results could be shown in the charts. The conclusions lack conclusions regarding the continuation of research and practical recommendations for public policies. A good assessment deserves quantitatively showing the essence of the relationship between financial development and greenhouse gas emissions. The article is worth publishing - it has scientific value.

Reviewer 2 Report

Based on panel data covering 30 provinces in China from 2000 to 2020, this paper employs a panel smooth transition regression (PSTR) model to examine the impact of financial development on carbon dioxide emissions. Some specific suggestions and comments are as follows.

1. There are a lot of grammatical errors in the paper, please correct them. For example, "groupe b" should be "Group b" in line 54.

2. line 54-54: Group a include Shanxi province, and Group b include Shanxi province. This is a contradiction.

3. Figure 1: The text description of the vertical axis of the coordinate is inappropriate.

4. The PSTR model was proposed by other authors, and the academic contribution of this paper is not clear.

5. Section 3.2: We did not find Table S1, Table S2 and Table S3 in the manuscript.

6. Section 1, Section 2, and Section 3 all have descriptions of the PSTR model that are a bit messy and require further integration of references.

7. Section 4: The value range of the parameter needs to be further explained.

8. The paper lacks a discussion section.

9. The conclusion of the paper is not thorough enough.

Reviewer 3 Report

The paper aims at employing the panel smooth transition regression (PSTR) model to examine the impact of financial development on carbon dioxide emissions.

Apart from some mathematics presented in the paper, the structure is weak as well as the research design. Conclusions and suggestions are not adequate and limited in their scope.

English language and overall format is really poor and several mistakes can be detected.

Reviewer 4 Report

In recent years, the development of industrialization and urbanization has had an impact on the environment, of which carbon dioxide emission is an important environmental pollution factor. The coordination of economic development and environmental protection is an urgent issue to be solved. This paper analyzes the relationship between economic development and carbon dioxide emissions through the PSTR model. This study investigates the advantages and disadvantages of existing models to study economic development and CO2 emission, describes the principle and improvement scheme of the PSTR model; sets up a linear experiment and a non-linear comparison experiment based on the CO2 emission data of 30 provinces during 2000-2020. The paper needs a revision as the presentation of the current manuscript is below the standard quality. Here are some suggestions:

1. There are many  in the text where the tense, singular and plural of words are improperly used, which are suggested to be revised, such as on line 39, "there is a tendency to makes sacrifices", the word "makes" is suggested to be used as "make ".

Line 40: "where economic development is pursued at the expense of the environment environmental improvement for people's survival and development is ignored." 

2. Several long sentences in the text are a bit vague, so we suggest using short sentences. The overall English presentation seems poor and requires a significant improvement.

3. There are a number of spelling errors throughout the manuscript, such as "Groupe b" in line 55. Please correct them.

4. All abbreviations should be given a full name before introducing them, e.g., SDA, IDA, DEA, GMM, etc.

5. The column "countries" in Table 1 does not contain the correct information, like the third and fourth rows.

Reviewer 5 Report

I find this research interesting. Still, the manuscript is poorly prepared, needs substantial improvements, and is not ready to be considered for publication in the present form. Here are my main comments and suggestions, which, if considered, I believe could help to improve the paper further.

(i) The Abstract is eclectic, and there is no link between its first and second parts. It is not clear why it is important to analyze the effect of financialization on GHG emissions since it is already an analyzed topic. If the novelty is related to the methodology, it is still not clear in the abstract what is wrong with the previous methodologies. What advantages PSTR has compared to other methods? What scientific problem paper analyzes? I think the Abstract needs a good revision.

(ii) All text should be reviewed to fix logical errors. For example, lines 79-81. It seems that Kuznec proposed EKC in 1955. It is not true since the original Kuznec curve links the country’s development level with inequality. The same inverted U-shaped form of relationship, but much later, was adopted to analyze the relationship between a country’s development level and environmental degradation and named EKC. There are sentences after which some information probably is missing (for example, line 265). 

(iii) In the Introduction section, the idea of the relationship between financial development and GHG emissions is underdeveloped. In the Literature review, the mechanism behind a negative correlation between financial development and environmental pollution is not deeply explained and grounded.

(iv) Table 1 should be moved to the literature review. Fig. 1 should be moved and discussed, presenting the data of the research. The Metodology must be explained more consistently. Now it is messy. For example, in line 282 appears ‘m’ which is not explained previously or after. In lines 290-291, the alternative models are not clear since variables used to proxy financial development are not discussed. Also, the transition function R, which appears in lines 321-322, is not previously developed. In general, data presentation and description in subsection 3.2. is absolutely inappropriate since no information is provided. No list of variables, no data, no descriptive statistics. Just references to nonexisting Tables S1 and S2.

(v) The size, placement and font of the equations and terms of the equations in the text are strange. They need some adjustment.